# Guidelines for Installation of Sensors in Smart Sensing Platforms in Underground Spaces

**DOI:** 10.3390/s22093215

**Published:** 2022-04-22

**Authors:** Zhenjiang Shen, Xiao Teng, Yuntian Zhang, Guoan Fang, Wei Xu

**Affiliations:** 1Fujian Science & Technology Innovation Laboratory for Optoelectronic Information of China, Fuzhou 350116, China; riendyteng@staff.kanazawa-u.ac.jp; 2Department of Environmental Science and Engineering, Fudan University, Shanghai 200433, China; 3Graduate School of Environment Design, Kanazawa University, Kanazawa 920-1192, Japan; 13991900799@163.com; 4Cross-Strait Tsinghua Research Institute, Fuzhou 361015, China; 5Shanghai Institute of Geological Survey, Shanghai 200072, China; tang.xiaoruy@gmail.com (G.F.); xuweishdz@gmail.com (W.X.)

**Keywords:** sensing system design, sensor parameters, underground space classification, data flow, design phase, construction phase, smart scenario

## Abstract

The purpose of this study is to propose guidelines for sensor installation in different types of underground space smart sensing platforms. Firstly, we classify the underground space, analyze the scene requirements according to the classification of underground space, and sort out the requirements for sensors in various types of underground space. Secondly, according to the requirements of underground space scenes for sensors, the types of sensors and corresponding parameters are clarified. After that, the system design and sensor installation guidelines of the underground space smart sensing platform are proposed by sorting out the data acquired by the sensor.

## 1. Introduction

This paper clarifies the sensor types and corresponding parameters for smart sensing scenes by analyzing the needs of different types of underground space scenes. Based on the sensor parameters and acquired data required by the underground space smart sensing platform system, the sensor installation guidelines are formed accordingly.

This study needs to clarify types of underground spaces, and we can refer to the laws and regulations of underground spaces in each country. Japan has a perfect legal and management mechanism for underground spaces [1,2,3,4], such as the “Act on Special Measures concerning Public Use of Deep Underground”, which clarifies the specific problems and technical measures for underground space utilization. Meanwhile, for different types of underground spaces, there are different technical standards and construction regulations [5], such as “Standard Specification for Tunneling” and so on. The “State Lands Act” and “Land Acquisition Act” clarify the classification basis and ownership of underground spaces, and the “Common Services Tunnels Act” and others have introduced new technical requirements for different underground spaces. There are also different technical standards and construction procedures in different underground space classifications such as “Railway applications Fixed installations Electric traction overhead contact lines”. In the United States, there is also a relatively complete legal and management mechanism for underground space [6], and state laws such as “Laws of Minnesota for 1985 Mined Underground Space” and “Oklahoma Statutes Property” have clarified the technical measures in the development and use of underground space according to the characteristics of each state. There are also different technical standards and construction procedures in different underground space types, such as “Underground Electric Distribution Standards Manual” and “Underground Construction (Tunneling)”. Similar to the United States, the United Kingdom also has a comprehensive legal and regulatory mechanism [6], such as the “London Underground Act 1992”, which proposes measures to deal with various problems in different areas of London’s underground space. The United Kingdom also has different technical standards and construction regulations in different underground space types, such as “The Road Tunnel Safety (Amendment) Regulations”. Through these laws and regulations, the classification of underground space can be clarified [7,8].

From the location of the underground space, it requires arterial energy from facilities such as water, electricity, transportation and data flow [9,10,11,12,13], and waste water and waste disposal through veins [13,14,15,16,17], while the underground space contains the infrastructure that ensures the function and operation of urban infrastructure and is the “lifeline” that combines the arteries and veins of the city [16,17,18,19,20,21,22] in addition to the services and facilities that bring benefits and taxes from commercial operations and provide commercial value to the city [23,24]. Therefore, the planning of underground spaces also emphasizes the actual equipment [25], and smart sensing of underground spaces can improve the responsiveness of equipment and thus the efficiency of the city’s arteries and veins [26,27]. As a basic component of smart construction, the installation of sensors (including model selection, location selection, combination mode, etc.) is an important part of smart work [28,29,30,31,32].

Based on the existing design guidelines in Japan, such as “Facility Construction Safety Construction Technical Guidelines” and “Civil Engineering Work Safety Construction Technical Guidelines”, it is clear in Table 1 that the requirements for equipment in underground spaces in the traditional Japanese design guidelines are mainly reflected in the types of equipment, the general location of equipment and the occasions of use of equipment. The existing regulations and guidelines basically do not involve smart devices, sensors and other new equipment content. This problem is common in the design guidelines for underground spaces in all countries. From the devices used in the current underground space in Japan, we can also see in Table 1 that these devices consume more energy and have a lower degree of smartness, and most of them are universal and do not select devices for the characteristics of different types of underground spaces.

In summary, countries have more complete laws and regulations for the infrastructure construction of underground space [33], but there is a lack of design guidelines for setting up smart sensing devices in the whole underground space [33]. Therefore, this study starts from the design phase of underground space, and after clarifying the classification of underground space, according to the functional requirements of different scenes for sensors in underground space, the type parameters of sensors in each scene are clarified. According to the type and parameters of sensors, the attributes of the collected data are clarified. Finally, based on the classification of underground space, the basic framework of underground space intelligent sensing system design, the properties of sensors, and the properties of data, we propose the guidelines of sensor installation for smart scenes in underground space.

## 2. Theoretical Concept and Methodology

To meet the monitoring and early warning needs of maintenance management, this study considers that the guidelines for setting up sensors in underground spaces need to clarify the classification of underground spaces and also the properties of sensors such as communication methods, the properties of data and the basic framework of the sensing system, thus taking the following research steps to propose guidelines for setting up sensors for smart scenes in underground spaces:Classify the underground space according to the classification standard of Japanese underground space and the functional characteristics of each type of underground space, and by clarifying the functional requirements of each type of underground space for sensors on the basis of the classification of underground space;Based on the above-mentioned requirements for sensors in the underground space, the sensors are screened on the basis of the temperature and humidity applicable to the underground space, and the sensor types and parameters are selected to meet the smart scene and functional requirements of the underground space;Based on the functional requirements met by the above sensors as well as the sensor types and parameters, the types of data acquired by each type of sensor and the data attributes are clarified;Based on the classification of underground space, sensor attributes and data attributes in I, II and III above, clarify the data transmission methods and data flow between sensors of various types of data, propose the basic framework of smart sensing system in underground space and form the guidelines for setting up sensors for smart scenes in underground space.

## 3. Underground Space Classification and Scene Requirements Analysis

Various types of underground spaces have different needs for sensors. Since the planning and construction of underground spaces in Japan is at an advanced level internationally, this study refers to the Japanese classification standard for underground space, which distinguishes between civilian land and public land, and classifies underground spaces according to the depth of various underground facilities in Figure 1.

Based on the Japanese classification standard for underground space, underground space can be divided into six types of underground infrastructure: rail transit, underground functional places (underground stores, parking lots, subway stations, etc.), elevated bridges, underground tunnels, underground municipal pipelines, and underground heat source heat pumps in Table 2.

Different functions of the underground space are monitored differently during maintenance. Therefore, there are also differences in the requirements for sensors. In order to meet the operational requirements (Figure 2), the discussion needs to be based on the classification of the underground space in question.

Underground spaces have different requirements for various types of sensors in different types of spaces, which are organized according to the Table 3 below.

## 4. Underground Space Sensor Sorting

Based on the classification of the underground space, and the requirements of sensors for different underground space scenarios, the sensors on the market are screened on the basis of the temperature and humidity applicable to underground space, and the types of sensors and the corresponding parameters of sensors that can meet the needs of the scenario are obtained in Table 4.

## 5. Underground Space Data Sorting

Based on the functional requirements of the underground space for sensors, sensor types and parameters, the attributes of the data acquired by various types of sensors are clarified, including data transmission methods and data monitoring scopes. The final results will be classified based on data types and attributes to form a data summary table (Table 5).

## 6. Basic Framework of Underground Space Smart Sensing Platform

Based on the analysis of underground space classification and scene requirements, sensor attributes and data attributes, the framework of underground space smart sensing platform (smart sensing system control) is proposed in Figure 3.

Based on the data transmission methods and data flow between sensors in the underground space, the data flow of the sensing platform is clarified in Figure 4: smart monitoring devices (sensors) acquire data Multiple sensors are combined to form smart scenarios The data acquired by each sensor is integrated as task (or the data acquired by each scene as Ambience), through the central processor and the database data for comparison, through the network transport module to the database module (update data without problems) or data report generation visualization real-time management module (problem data for feedback) feedback to smart monitoring equipment (alarm device) through forecast and early warning module with problematic data.

## 7. Production of Sensor Installation Guidelines for Underground Space

In this paper, we make a summary analysis of the requirements of different underground space types to clarify the types of underground space sensor requirements as well as the parameters of the sensors and the monitoring data of the underground space to form the sensor installation guidelines in the underground space smart sensing platform. To facilitate the designers to carry out the design work related to the underground space smart, we will finally form the sensor design and installation guidelines table (Table 6) and the installation location schematic according to the characteristics of the sensors used in different scenarios (Figure 5, Figure 6, Figure 7, Figure 8, Figure 9, Figure 10, Figure 11, Figure 12, Figure 13, Figure 14, Figure 15 and Figure 16).

## 8. Conclusions

In this paper, we analyze the requirements of different types of underground space scenes and clearly establish the sensor types and corresponding parameters for smart sensing scenes. Based on the sensor parameters and the acquired data, we propose a system design for smart sensing platform in underground space and form the sensor installation guidelines accordingly. The guidelines are mainly divided into two parts: installation guideline table and installation schematic, and the purpose of setting the guidelines is to guide the work related to the construction of smart scenes in underground space.

The study of requirements in this paper focuses largely on maintenance management. The next step will be to refine the process of developing and constructing smart scenes in underground spaces, and to propose different development proposals and guidelines at different phases according to the refined process.

## Figures and Tables

**Figure 1 sensors-22-03215-f001:**
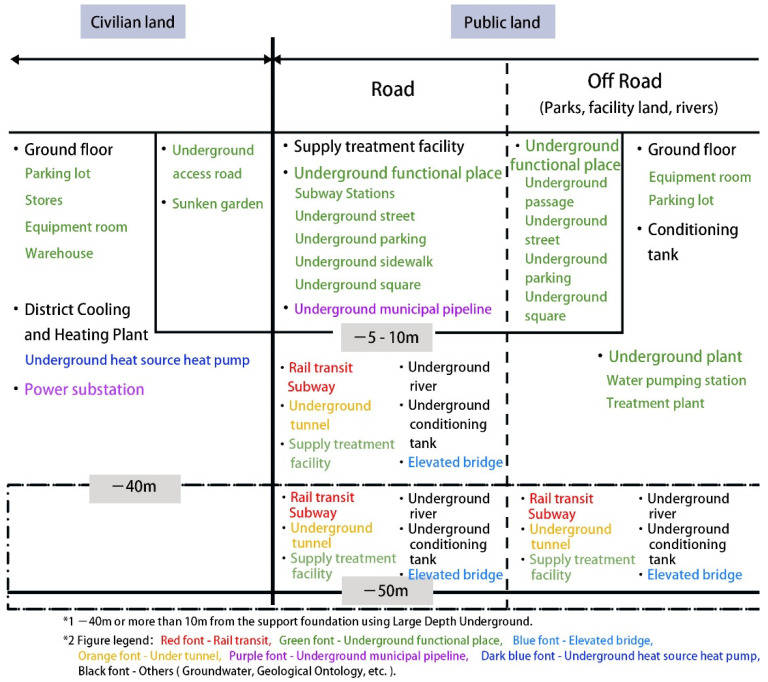
Classification of underground space in Japan.

**Figure 2 sensors-22-03215-f002:**
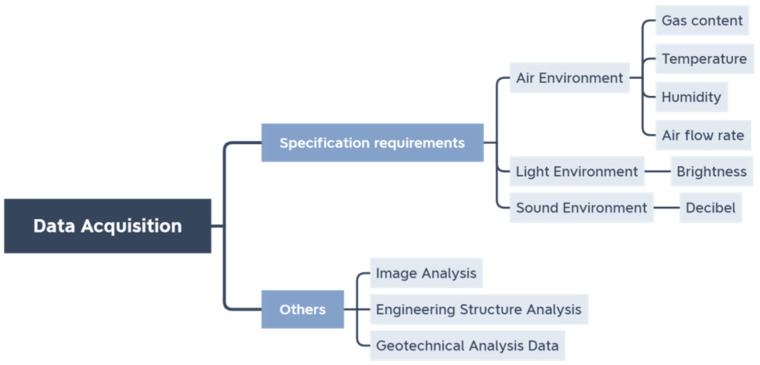
Maintenance phase monitoring content classification diagram.

**Figure 3 sensors-22-03215-f003:**
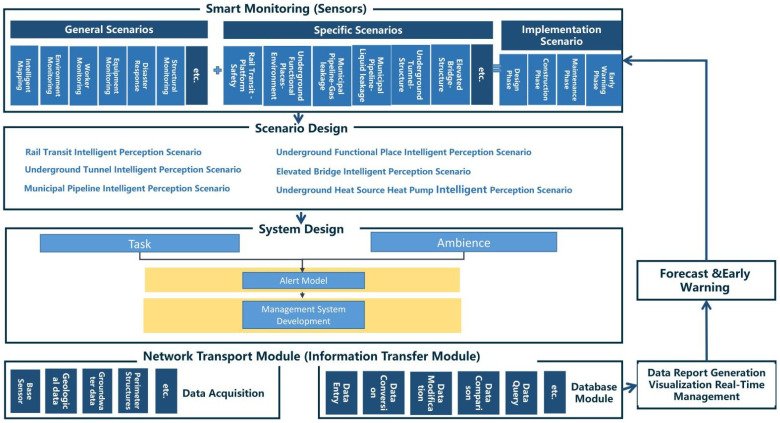
Basic framework of underground space smart sensing platform.

**Figure 4 sensors-22-03215-f004:**
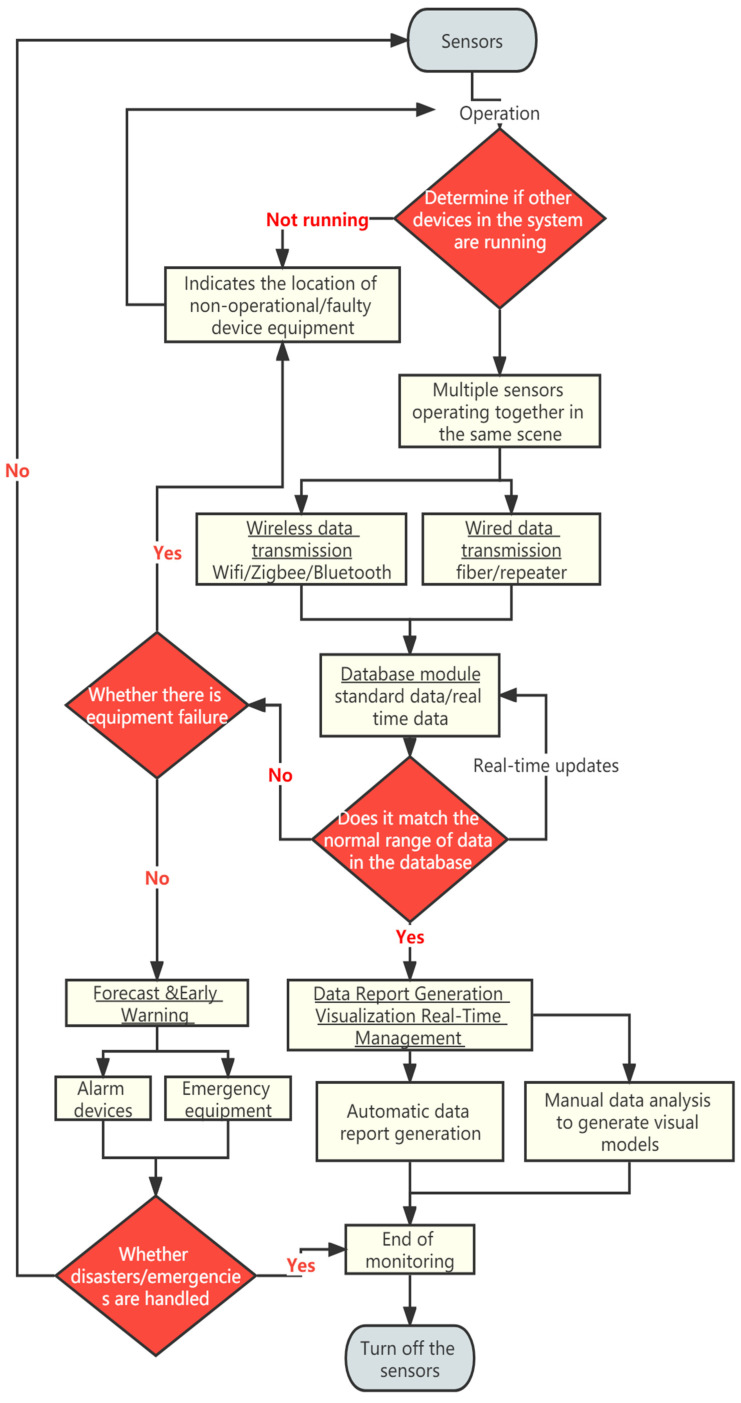
Smart perception platform UML.

**Figure 5 sensors-22-03215-f005:**
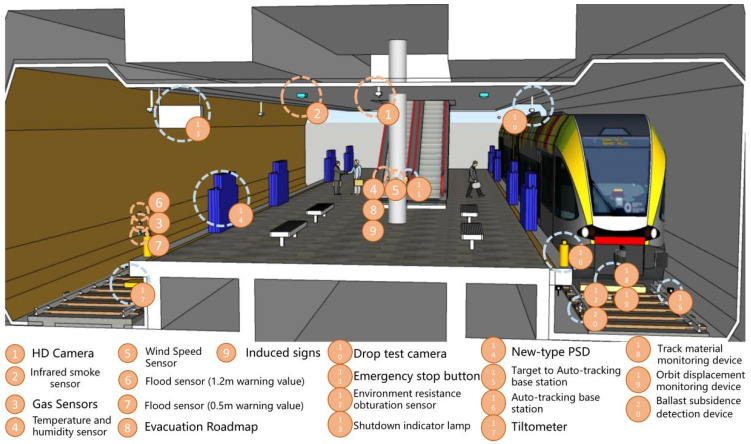
Rail transit sensor installation schematic.

**Figure 6 sensors-22-03215-f006:**
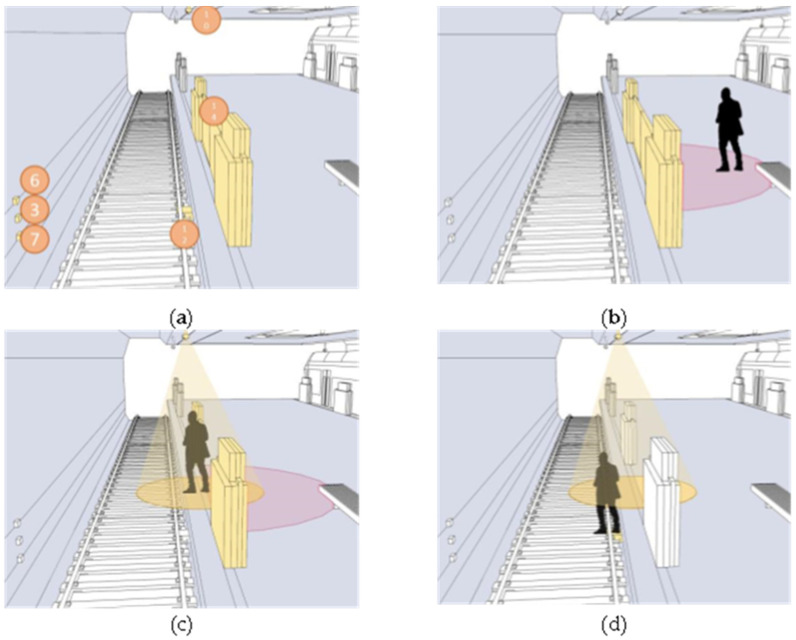
Smart interaction function for rail transit. (**a**): Various types of devices in smart interaction scenarios. Including environmental monitoring devices and accident response devices. (**b**): User access to the detection range of new-type PSD. (**c**): The door opens automatically after entering the detection range of new-type PSD, after which drop test camera opens and the user enters the detection range of drop test camera. (**d**): When the user falls off the track, environment resistance obturation sensor will sound an alarm. He or she will remain in the detection range of drop test camera. At this time, drop test camera will determine the location of the fall accident and upload it.

**Figure 7 sensors-22-03215-f007:**
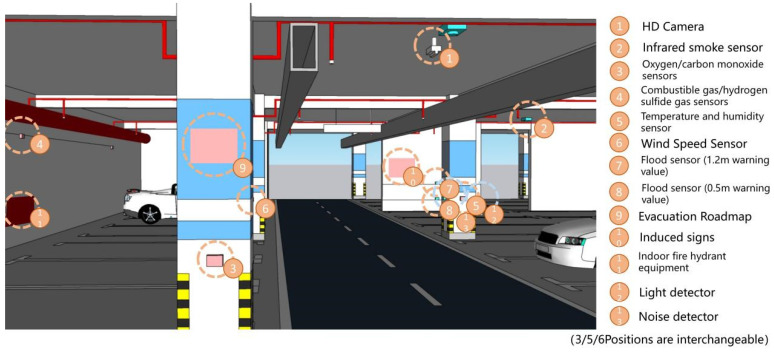
Underground functional place sensor installation schematic.

**Figure 8 sensors-22-03215-f008:**
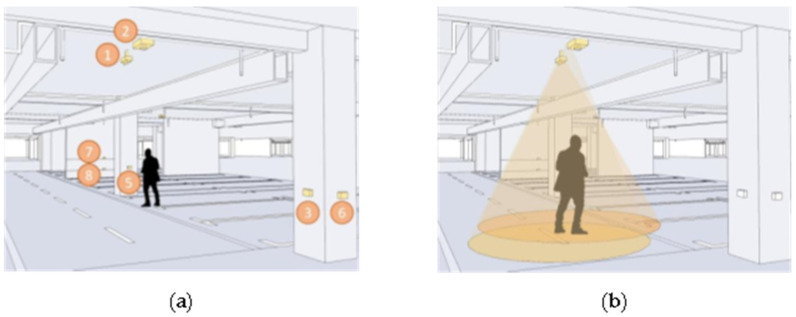
Smart interaction function for underground functional place. (**a**): Various types of devices in smart interaction scenarios. Environmental monitoring equipment is the main focus. Users can read basic environmental information at temperature and humidity sensor. (**b**): When a user is detected to be in the monitoring range, the devices mainly in HD camera are automatically turned on.

**Figure 9 sensors-22-03215-f009:**
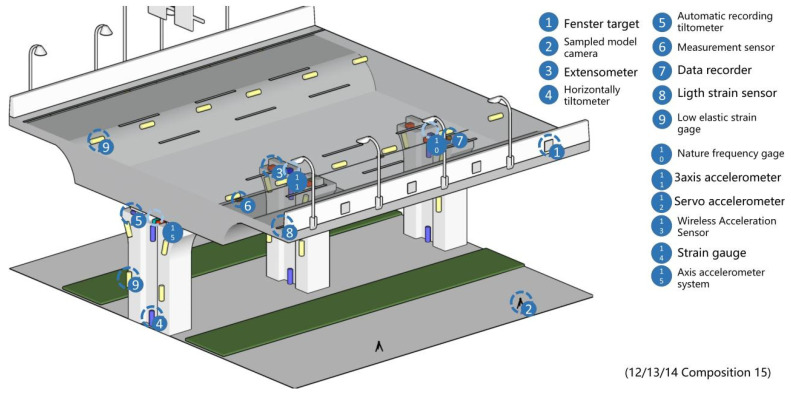
Elevated bridge sensor installation schematic.

**Figure 10 sensors-22-03215-f010:**
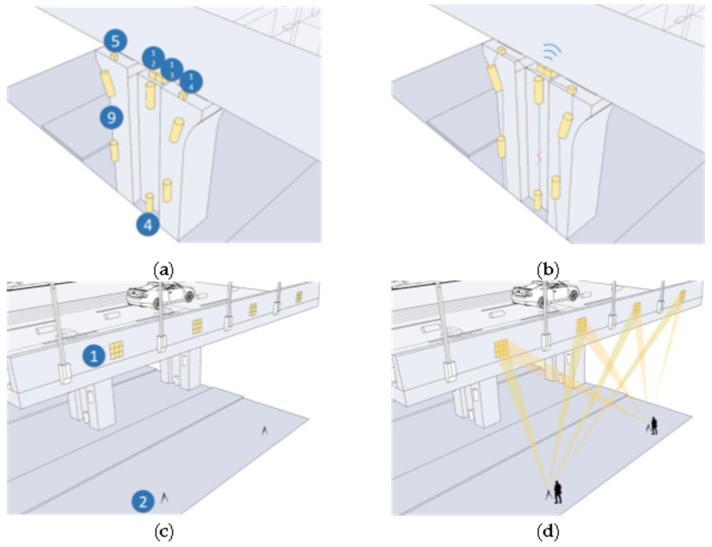
Smart interaction function for elevated bridge. (**a**): The monitoring devices are mainly column structure monitoring devices. (**b**): The various devices are linked by optical fibers, and when cracks appear/structure damage, electrical signals are conducted from measurement sensors/horizontally tiltometer to 3-axis accelerometer system and uploaded by 3-axis accelerometer system via optical fibers. (**c**): Bridge structure monitoring equipment is the main focus. (**d**): The user captures the location of multiple fenster target’s by sampled model camera and determines if the bridge is deformed by comparing it to the indicator.

**Figure 11 sensors-22-03215-f011:**
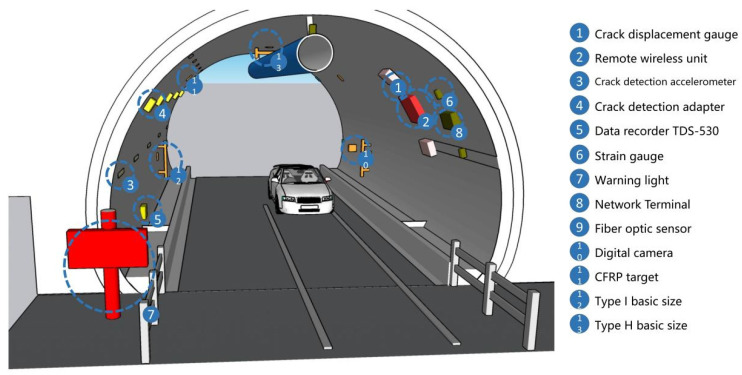
Underground tunnel sensor installation schematic.

**Figure 12 sensors-22-03215-f012:**
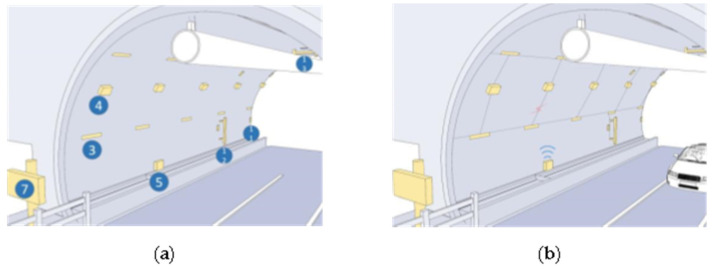
Smart interaction function for underground tunnel. (**a**): The smart devices are mainly tunnel support structure monitoring devices. (**b**): Various devices are linked by fiber optics. When a crack/structure breakage occurs in the support, the electrical signal from the sensor is stored to data recorder via optical fiber and uploaded by data recorder.

**Figure 13 sensors-22-03215-f013:**
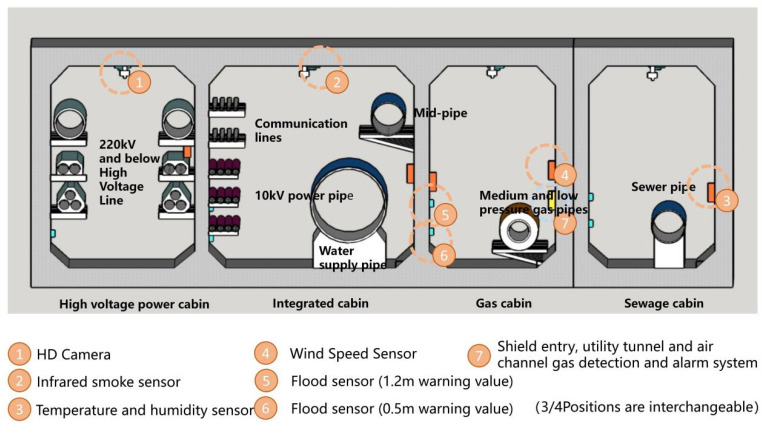
Underground municipal pipeline sensor installation schematic.

**Figure 14 sensors-22-03215-f014:**
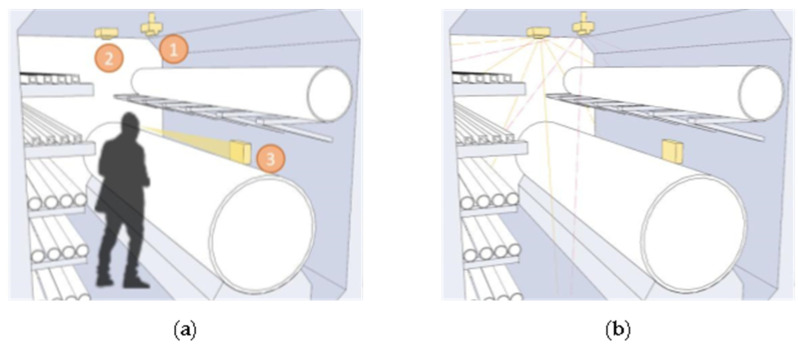
Smart interaction function for underground municipal pipeline. (**a**): The smart devices are divided into environmental monitoring devices and pipe corridor structure monitoring devices. Users can read basic environmental information from temperature and humidity sensor on the walls of the corridor. (**b**): The monitoring range of environmental monitoring devices covers the entire corridor space.

**Figure 15 sensors-22-03215-f015:**
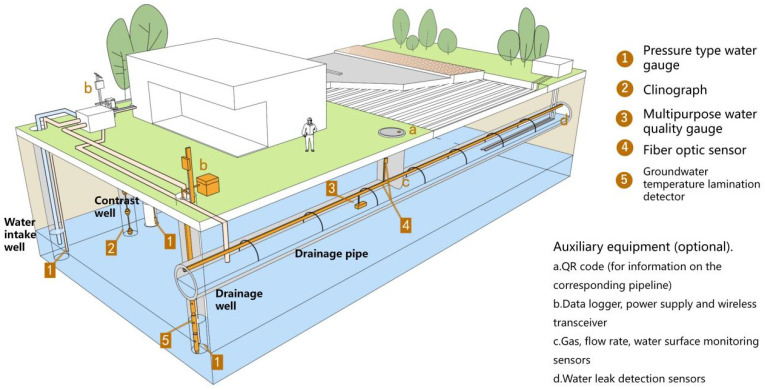
Underground heat source heat pump sensor installation schematic.

**Figure 16 sensors-22-03215-f016:**
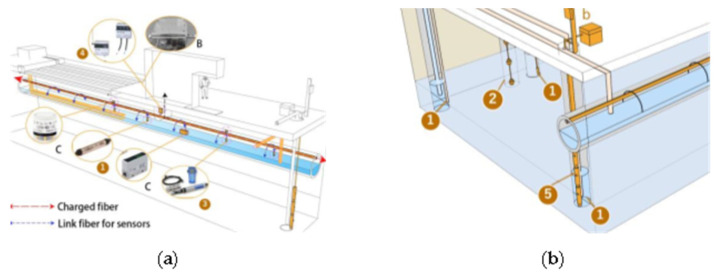
Smart interaction function for underground heat source heat pump. (**a**): Smart devices in drainage pipes. Water monitoring devices and pipeline structure monitoring devices are the main focus. All kinds of devices are linked by optical fiber, and the electrical signal is transmitted to the data recorder and wireless transceiver after water quality problems or structural problems, and uploaded by the wireless transceiver. (**b**): Smart devices in comparison wells, intake wells and drainage wells. Water quality monitoring devices and well structure monitoring devices are the main focus.

**Table 1 sensors-22-03215-t001:** Summary of the requirements of the existing guidelines for underground space devices and the devices commonly used in the current underground space.

The Types of Equipment Mentioned in the Guidelines	Rules for Equipment Requirements Mentioned in the Guidelines	Current Use of the Devices
Lighting Equipment	1. Requirements on the installation of lighting equipment in three cases: a. Lighting equipment renewal (completely renewed, mixed old and new); b. Lighting equipment operation next to the air exchange fan. 2. Precautions for wiring installation of lighting equipment. 3. Precautions for installation of lighting equipment itself.	LED tunnel light
Air Ventilation Equipment	1. The air exchange equipment is mainly tunnel fans. 2. The location of the anchor for the fan is determined and installation precautions. 3. Precautions for installation of the fan itself. 4. Post-installation testing precautions.	Tunnel jet fan
Dust Countermeasure Equipment	1. Application of air ventilation equipment in the dust response phase. 2. Specific content of dust concentration measurement. 3. Precautions for the use of respiratory protective equipment in emergency situations.	Tunnel jet fan, dust concentration detector
Noxious Gas Response Equipment	1. Requirements for exhaust devices. 2. Requirements for alarm devices (including implementation of monitoring equipment). 3. Response when the critical value is reached (automatic power cut). 4. Requirements for evacuation equipment.	Tunnel jet fans, alarms or emergency bells, Automatic fixed combustible gas alarms, Automatic power cut-off devices
Alarm & Rescue Equipment	1. Clarify the investigation content of the prior investigation. 2. The use of rescue equipment. 3. The use of alarm equipment.	Alarm or emergency bell, automatic fixed combustible gas alarm, smoke detector
Disaster Response Equipment	To deal with rain, wind, snow, lightning, earthquakes and other natural disasters	Alarm or emergency bell, automatic fixed combustible gas alarm, automatic power cut-off device
Structure Monitoring Equipment	Mainly construction auxiliary equipment.	Earth pressure meter (vibrating string type earth pressure meter), pore water pressure meter, static level, displacement meter, in-clinometer, pillar pressure meter, reinforcement meter/anchor force gauge, concrete strain gauge
Environmental Monitoring Equipment	1. Measures for places with poor ventilation conditions. 2. Measures to deal with the cramped environment during mechanical construction. 3. Measurement required for operating environment.	Automatic fixed combustible gas alarm, smoke detector, thermometer, hygrometer, dust concentration detector

Source: compiled based on Facility Construction Safety Construction Technical Guidelines and Civil Engineering Work Safety Construction Technical Guidelines.

**Table 2 sensors-22-03215-t002:** Classification of underground space and monitoring priorities.

Scene Type	Maintenance and Management		Control-Warning
	Daily Maintenance	Disaster Prevention	
Rail transit	Personal safety-fall, Personal safety-attention wake-up, obstacle monitoring, track status monitoring	Flooding disaster, fire evacuation, earthquake disaster	smart construction monitoring, disaster response (fire, flooding, earthquake, gas leak, tilt, subsidence, deformation)
Elevated bridge	Structural security monitoring		
Underground tunnel	Structural security monitoring	Tube sheet disease, tunnel flooding disaster, fire evacuation, earthquake disaster	
Underground municipal pipeline	Pipeline structure monitoring, pipe chamber environmental monitoring	Pipeline leaks (liquid/gas), underground voids	
Underground functional place	Underground space environmental monitoring	Underground space flooding disaster, fire evacuation, earthquake disaster	
Underground heat source heat pump	Underground heat source pollution monitoring		
Geological ontology	Surface monitoring, in-ground monitoring		Geological tilt, subsidence, deformation
Ground water	Groundwater level daily monitoring	Groundwater contamination monitoring	Groundwater level, water temperature, water quality abnormalities

Source: compiled based on Act on Special Measures Concerning Public Use of Deep Underground.

**Table 3 sensors-22-03215-t003:** Summary of underground space requirements.

Smart Scene Type	Requirements	Purpose
Basic scenes (generic scenes)	Monitoring of natural and man-made disasters	Response before and after disasters, mainly natural disasters (flooding, earthquakes, extreme weather (extreme cold and heat, thunderstorms)) and man-made disasters (fire, equipment failure, construction personnel health)
	Monitor construction and operational environments	Monitoring and adjusting the air environment, sound environment, geotechnical environment, and light environment to make people feel comfortable
	Obtaining staff health information	Prevention of various types of emergencies in the underground space when affecting the health of staff, timely response
	Obtain equipment movement information	Prevent the loss of equipment in the underground space or loss of contact with the host
Rail transit	Obtain information on falling objects on the tracks	Prevent people from falling onto the track or objects from falling onto the track and affecting train operation
	Monitoring of platform doors for objects caught in them	Prevent damage to people or objects when platform doors are closed
	Emergency stop of trains	To guide trains to an emergency stop after an emergency situation to reduce damage
	Obtain track structure information	Prevent damage to the track structure from affecting operation
Underground functional place	Obtain operating environment data and make timely adjustments	To make the environment comfortable and convenient for various functions
Elevated bridge	Obtain bridge structure information	Reducing damage to bridge structures that may affect operations
	Obtain information on pillar structure	Prevention of collapse due to damage to the column structure
Underground tunnel	Obtain support structure information	Reduce the collapse potential caused by damage such as support cracks, and deal with cracks that have a greater impact in a timely manner
Underground municipal pipeline	Real-time monitoring of pipeline structure	Prevent the occurrence of liquid leakage and air leakage, and respond to liquid leakage and air leakage in a timely manner
Underground heat source heat pump	Obtain information on pipeline structure	Prevent fluid leaks and respond to them in a timely manner
	Obtain information on the structure of the exchange well	Reduce collapse hazards caused by cracks in exchange wells, etc., and promptly respond to cracks that have a large impact
	Monitoring the fluid in the well	Liquid temperature and water level should meet the specific requirements of the heat source heat pump and reduce the influence of the liquid itself on the efficiency of the heat source heat pump
	Obtain information on surrounding groundwater	Reduce the pollution of the surrounding groundwater by the liquid in the exchange well

Source: Compiled based on documents related to underground space issued by the Ministry of Land, Infrastructure, Transport and Tourism (MLIT).

**Table 4 sensors-22-03215-t004:** Summary of underground space sensors.

Smart Scene Type	Sensor Type	Detection Principle (Component Equipment)	Data Monitoring Scope	Long-Term/Regular Monitoring
Basic scenes (generic scenes)	Infrared smoke sensor	Infrared 2 wavelength type, fluctuation type, CO_2_ resonance radiation type	Exposure to smoke YES/NO	Long-term monitoring
	Gas sensors	Hot-wire type semiconductor type, contact combustion type, gas heat conduction type	0~100% LEL	Long-term monitoring
	Hydrogen sulfide sensor		0~50 ppm	Long-term monitoring
	Carbon monoxide sensor	Constant potential electrolytic type, diaphragm plus Varney cell type	0~250 ppm	Long-term monitoring
	Oxygen sensor	Constant potential electrolytic type, diaphragm plus Varney cell type	0~25 vol%	Long-term monitoring
	Noise detector1	Condenser electric microphone	28~141 dB	Long-term monitoring
	Flood sensor	Water contact sensor	Exposure to water YES/NO	Long-term monitoring
	Temperature and humidity sensor	Capacities temperature and humidity sensitive sensor	Temperature: −40~80 °C Humidity: 0~100%Rh	Long-term monitoring
	Mobile environmental monitoring sensor	Constant potential electrolytic type, diaphragm plus Varney cell type	Oxygen concentration: 0~25 vol%	Long-term monitoring
			Carbon monoxide concentration: 0~300 ppm	Long-term monitoring
	Laser distance sensor	Light reflection principle	0.05~100 m	Long-term monitoring
	RFID readers/tags	Light reflection principle	3–5 m	Long-term monitoring
	Dust sensor	Light scattering relative density meter	Exposure to dust YES/NO	Long-term monitoring
	Wind speed sensor	Rotor rotation speed	0~114 m/s	Long-term monitoring
Rail transit	Gas sensors	Hot-wire type semiconductor type, contact combustion type, gas heat conduction type, constant potential electrolytic type, diaphragm plus Varney cell type	0~100% LEL	Long-term monitoring
	Hydrogen sulfide sensor		0~50 ppm	Long-term monitoring
	Carbon monoxide sensor		0~250 ppm	Long-term monitoring
	Oxygen sensor		0~25 vol%	Long-term monitoring
	Construction monitoring radar	Light reflection principle	Collapse occurs YES/NO	Long-term monitoring
	Roll-off detection mat	Environment resistance obturation sensor, Drop test camera	Perceived pressure YES/NO	Long-term monitoring
	New-type PSD	Residual detection sensors (3D sensors, photoelectric sensors), proximity detection sensors (photoelectric sensors)	Perceived pressure YES/NO	Long-term monitoring
	Track material monitoring device	Configuration camera (distance image capture device)	Material Breakage YES/NO	Regular monitoring
	Orbit displacement monitoring device	Linear sensor camera (Intense and faint image photography device)	Orbital displacement YES/NO	Regular monitoring
	Image displacement measurement system	Laser displacement meter	Orbital displacement YES/NO	Regular monitoring
	Line equipment monitoring device	Digital camera, displacement meter, in-clinometer	Equipment breakage YES/NO	Regular monitoring
Underground functional place	Dust sensor	Light scattering relative density meter	Exposure to dust YES/NO	Long-term monitoring
Elevated bridge	Extensometer	Displacement sensor	6.5 ± 1 mm	Long-term monitoring
	Horizontally tiltometer	Tiltmeter	0–500 mm	Long-term monitoring
	Crack gauge	Crack gauge	5~40%	Long-term monitoring
	Light strain sensor	Light strain sensor, Strain gauge, Fiber optic measuring instrument	Sensing to strain YES/NO	Long-term monitoring
	Nature frequency gage	Nature frequency gage	50 kHz	Regular monitoring
Underground tunnel	Gas sensors	Hot-wire type semiconductor type, contact combustion type, gas heat conduction type, constant potential electrolytic type, diaphragm plus Varney cell type	0~100% LEL	Long-term monitoring
	Hydrogen sulfide sensor		0~50 ppm	Long-term monitoring
	Carbon monoxide sensor		0~250 ppm	Long-term monitoring
	Oxygen sensor		0~25 vol%	Long-term monitoring
	Construction monitoring radar	Light reflection principle	Collapse occurs YES/NO	Long-term monitoring
	Crack displacement meter	Crack displacement meter, remote wireless unit	5~40%	Long-term monitoring
	Fiber optic crack detection sensor	Crack detection accelerometer, crack detection adapter, data recorder TDS-530	5~40%	Long-term monitoring
	Fiber optic sensor	Fiber optic sensor	Cracking occurs YES/NO	Long-term monitoring
Underground municipal pipeline	Water leak detection service	Water leak detection sensor	Water leakage occurs YES/NO	Long-term monitoring
	Remote water leak monitoring system	Water leak detection sensor	Water leakage occurs YES/NO	Long-term monitoring
	Installation of tube lumen survey machine	Electromagnetic pulse radar, television cameras	Tube lumen breakage YES/NO	Regular monitoring
Underground heat source heat pump	Pressure type water gauge	Induction of hydro-static pressure in water bodies	0.05% F.S	Long-term monitoring
	Clinograph	Tilt sensor, electrolyte and conductive contacts	±330 micro-radius	Long-term monitoring
	Multipurpose water quality gauge	Voltage conductivity	PH value	Long-term monitoring
	Water leak detection sensor	Laser hydrostatic principle	0–50 m	Long-term monitoring

Source: Compiled based on the public information of the company: Tokyo Measuring Instruments Lab. (Kiryu Factory, Kiryu, Japan); SAKATA DENKI Co., Ltd. (Head Office & Factory, Tokyo, Japan); New Cosmos Electric Co., Ltd. (Cosmos sensor Center, Hyogo, Japan; Tokyo Factory, Tokyo, Japan); AIREC ENGINEERING Corporation (Head Office & Factory, Tokyo, Japan); TOBISHIMA Corporation, Keyence Co., Ltd. (Head Office & Factory, Tokyo, Japan); Kyosan Electric Mfg. Co., Ltd. (Head Office & Factory, Yokohama, Japan; Zama Factory, Kanagawa, Japan); Japan Railway Track Consultants Co., Ltd. (Head Office & Factory, Tokyo, Japan); Kyowa Electronic Instruments Co., Ltd. (Kofu Kyowa Dengyo Co., Ltd., Yamanashi, Japan; Yamagata Kyowa Dengyo Co., Ltd., Yamagata, Japan) where each type of sensor is located.

**Table 5 sensors-22-03215-t005:** Sensor-based data summarization in underground spaces.

Smart Scene Type	Sensor Type	Data Monitoring Scope	Monitoring Indicator	Data Transmission Method
Basic scenes (generic scenes)	Infrared smoke sensor	Exposure to smoke YES/NO	Smoke, thermal infrared	ZigBee/Bluetooth
	Gas sensors	0~100% LEL	Hydrogen concentration, sulfur dioxide gas concentration, carbon dioxide gas concentration	ZigBee/Bluetooth
	Hydrogen sulfide sensor	0~50 ppm	Hydrogen sulfide gas concentration	ZigBee/Bluetooth
	Carbon monoxide sensor	0~250 ppm	Carbon monoxide gas concentration	ZigBee/Bluetooth
	Oxygen sensor	0~25 vol%	Oxygen concentration	ZigBee/Bluetooth
	Noise detector	28~141 dB	Noise intensity	WiFi/Bluetooth
	Flood sensor	Exposure to water YES/NO	Flooding depth	WiFi/Bluetooth
	Temperature and humidity sensor	Temperature: −40~80 ℃ Humidity: 0~100% Rh	Temperature & Humidity	WiFi/Bluetooth
	Mobile environmental monitoring sensor	Oxygen concentration: 0~25 vol%	Oxygen concentration, Carbon monoxide gas concentration	WiFi/Repeater/Bluetooth
		Carbon monoxide concentration: 0~300 ppm		
	Laser distance sensor	0.05~100 m	Distance of mobile devices from the perimeter	Repeater-Bluetooth
	RFID readers/tags	3–5 m	Location, trajectory	WiFi/Bluetooth/USB
	Dust sensor	Exposure to dust YES/NO	Dust concentration (PM2.5 mainly)	WiFi/Repeater/Bluetooth
	Dust sensor	Exposure to dust YES/NO	Dust concentration	WiFi/Repeater/Bluetooth
	Wind speed sensor	0~114 m/s	Wind speed	WiFi/Repeater/Bluetooth
Rail transit	Gas sensors	0~100% LEL	Hydrogen concentration, sulfur dioxide gas concentration, carbon dioxide gas concentration	WiFi/Bluetooth
	Hydrogen sulfide sensor	0~50 ppm	Hydrogen sulfide gas concentration	
	Carbon monoxide sensor	0~250 ppm	Carbon monoxide gas concentration	
	Oxygen sensor	0~25 vol%	Oxygen concentration	
	Construction monitoring radar	Collapse occurs YES/NO	Construction safety (construction environment)	Repeater-Bluetooth
	Roll-off detection mat	Perceived pressure YES/NO	Orbital drop	Fiber optic
	New-type PSD	Perceived pressure YES/NO	Rail platform gap	Fiber optic
	Track material monitoring device	Material Breakage YES/NO	Track material	Repeater
	Orbit displacement monitoring device	Orbital displacement YES/NO	Orbital displacement distance	Repeater
	Image displacement measurement system	Orbital displacement YES/NO	Orbital displacement distance	Repeater
	Line equipment monitoring device	Equipment breakage YES/NO	Wires on the track	Repeater
Underground functional place	Dust sensor	Exposure to dust YES/NO	Dust concentration	WiFi/Repeater/Bluetooth
Elevated bridge	Extensometer	6.5 ± 1 mm	Bridge support displacement distance	Repeater-WiFi
	Horizontally tiltometer	0–500 mm	Inclined amount of bridge	Repeater-WiFi
	Crack gauge	5~40%	Crack width of bridge body	Repeater-WiFi
	Light strain sensor	Sensing to strain YES/NO	Bridge strain variables	Repeater-WiFi
	Nature frequency gage	50 kHz	Vibration characteristics of the bridge	Repeater-WiFi
Underground tunnel	Gas sensors	0~100% LEL	Hydrogen concentration, sulfur dioxide gas concentration, carbon dioxide gas concentration	WiFi/Bluetooth
	Hydrogen sulfide sensor	0~50 ppm	Hydrogen sulfide gas concentration	
	Carbon monoxide sensor	0~250 ppm	Carbon monoxide gas concentration	
	Oxygen sensor	0~25 vol%	Oxygen concentration	
	Construction monitoring radar	Collapse occurs YES/NO	Construction safety (construction environment)	Repeater-Bluetooth
	Crack displacement meter	5~40%	Tunnel support cracks	Repeater-WiFi
	Fiber optic crack detection sensor	5~40%	Tunnel support cracks	Repeater-WiFi
	Fiber optic sensor	Cracking occurs YES/NO	Tunnel support strain variables	Repeater-WiFi
Underground municipal pipeline	Water leak detection service	Water leakage occurs YES/NO	Pipeline liquid leakage	Repeater-WiFi
	Remote water leak monitoring system	Water leakage occurs YES/NO	Pipeline liquid leakage	Repeater-WiFi
	Installation of tube lumen survey machine	Tube lumen breakage YES/NO	Pipeline structure	Repeater-WiFi
Underground heat source heat pump	Pressure type water gauge	0.05% F.S	Heat source heat pump feed well water level, water temperature	Repeater-WiFi
	Clinograph	±330 micro-radius	Sliding surface depth, sliding direction and movement of heat source heat pump feeder wells	Repeater-WiFi
	Multipurpose water quality gauge	PH value	Water quality changes in heat source heat pump drainage wells	Repeater-WiFi
	Water leak detection sensor	0–50 m	Location and flow conditions of groundwater fluidized bed of heat source heat pump	Repeater-WiFi

Source: Compiled based on the public information of the company: Tokyo Measuring Instruments Lab. (Kiryu Factory, Kiryu, Japan); SAKATA DENKI Co., Ltd. (Head Office & Factory, Tokyo, Japan); New Cosmos Electric Co., Ltd. (Cosmos sensor Center, Hyogo, Japan; Tokyo Factory, Tokyo, Japan); AIREC ENGINEERING Corporation (Head Office & Factory, Tokyo, Japan); TOBISHIMA Corporation, Keyence Co., Ltd. (Head Office & Factory, Tokyo, Japan); Kyosan Electric Mfg. Co., Ltd. (Head Office & Factory, Yokohama, Japan; Zama Factory, Kanagawa, Japan); Japan Railway Track Consultants Co., Ltd. (Head Office & Factory, Tokyo, Japan); Kyowa Electronic Instruments Co., Ltd. (Kofu Kyowa Dengyo Co., Ltd., Yamanashi, Japan; Yamagata Kyowa Dengyo Co., Ltd., Yamagata, Japan) where each type of sensor is located.

**Table 6 sensors-22-03215-t006:** Sensor installation guidelines.

Smart Scene Type	Smart Scene Function	Sensor Type	Suitable Installation Location	Installation Method	Testing Requirements
Basic scenes (generic scenes)	Real-time monitoring of the operating and construction environment	Gas sensors	Support left and right wall, place left and right wall/column	Excavation: the amount of excavated soil and the amount of soil transportation (earth calculation) need to be clarified; Retaining support construction: thoroughly check excavation depth, soil quality, groundwater level, working soil pressure, etc., including installation of measuring equipment.	Earth calculation after excavation in construction (excavation volume, discharge and excavation volume, construction progress (excavation depth)); Trial run: power test, lighting test, various equipment operation test, environmental suitability test.
		Hydrogen sulfide sensor	Support left and right wall, place left and right wall/column		
		Carbon monoxide sensor	Support left and right wall, place left and right wall/column		
		Oxygen sensor	Support left and right wall, place left and right wall/column		
		Noise detector	Support left and right wall, place left and right wall/column		
		Temperature and humidity sensor	Support left and right wall, place left and right wall/column		
		Mobile environmental monitoring sensor	Staff members wear them everywhere		
		Dust sensor	Support left and right wall, place left and right wall/column, construction site floor		
		Wind speed sensor	Support left and right wall, place left and right wall/column		
	Monitor all kinds of emergencies	Flood sensor	Vertical safety distance of support/side wall from the ground		
		Infrared smoke sensor	Top of support, top of place safety distance of support/side wall from the ground		
	Get device movement information	Laser distance sensor	Placement with mobile devices		
		RFID readers/tags	Readers: Placement with mobile devices; Tags: Top of support/place		
Rail transit	Real-time construction monitoring	Construction monitoring radar	Ground (near construction site)	Underground diaphragm wall method: prevention of excavation wall collapse, attention to the construction environment geotechnical structure. Weathervane work method: after the structure confirms the foundation support by foundation endurance test, it fills the concrete filled in the working chamber in a dry environment. Shield construction method: excavation is carried out using an excavator, and then a block called a section is installed on the wall to construct a tunnel, and the excavation and discharge of sand and soil is carried out continuously.	1. Electricity test after the dentsu project (electric communication security); 2. Track commissioning test.
	Response to falling rail accidents	Roll-off detection mat	Both sides of the track		
		Drop test camera	Top of the wall directly above the platform door		
	Response to platform door accidents	New-type PSD	Both sides of the platform near the train		
	Orbital structure information acquisition	Track material monitoring device	Mounted with the bottom of the train		
		Orbit displacement monitoring device	Mounted with the bottom of the train		
		Image displacement measurement system	Mounted with the bottom of the train		
		Line equipment monitoring device	Mounted with the bottom of the train		
Underground functional place	Obtain and adjust operational environment data	Dust sensor	The left and right walls/columns of the place, the ground of the personnel gathering area can be placed separately	Same as basic scenes	Earthwork calculation after excavation; Trial run is based on environmental suitability test. There are differences in the requirements of the commissioning equipment according to the function.
Elevated bridge	Obtain bridge structure information	Extensometer	Contact part between column and bridge body	Ground drilling method: mainly soft foundation. As a general local piling construction method is the auger construction method; Shell method: mainly hard foundation. Swing and press into the outer cover hose within the full length of the pile. Mainly need to prevent the foundation from collapse.	Same as basic scenes
		Horizontally tiltometer	Mounted on the column		
		Crack gauge	Vulnerable points on columns/bridge deck		
		Light strain sensor	Bridge side (side wall)		
		Nature frequency gage	Contact part between column and bridge body		
Underground tunnel	Real-time construction monitoring	Construction monitoring radar	Ground (near construction site)	Shield construction method: continuous excavation and discharge of sand and soil is required; Earth cutting: to prevent subsidence, groundwater protrusion and inflow of soil and sand into the end well, reinforcement and improvement of the soil around the cavern ring are required; Soil cutting volume: the cutting soil and sand must be discharged exactly in line with the amount of excavation; 4. Equipment: the shield machine has the feature that it can only enter but not retreat, so pay attention to the construction status of the shield machine.	Same as basic scenes
	Obtain information on support structures	Crack displacement meter	Support sidewalls (near sidewall lines)		
		Fiber optic crack detection sensor	Support sidewalls (near sidewall lines)		
		Fiber optic sensor	Support sidewalls (near sidewall lines)		
Underground municipal pipeline	Real-time monitoring of pipeline structures	Water leak detection service	Liquid pipeline vulnerability point (turning point)	For PC grouting, grout mixers, grout pumps, flow meters (grout flow meters) and, in some cases, grouting equipment are used.	There are various tests such as PC grouting temperature measurement, chloride ion content, compressive strength, archeology test, etc.
		Remote water leak monitoring system	Liquid pipeline vulnerability point (turning point)		
		Installation of tube lumen survey machine	Inside the liquid pipeline		
Underground heat source heat pump	Obtain information about the structure in the exchange well	Clinograph	Exchange well interior side wall	1. Pipeline part. ① Confirmation of pipeline paths for misconnection. ② checking the depth of buried pipeline; ③ implementing water pressure test. ④ laying of buried marker plate. ⑤ setting buried markers on the ground surface. ⑥ Confirming the construction around the header. 2. Heat exchange well section. ① Capture geological information Record in excavation. ② Simultaneous setting confirmation of water tension in the underground heat ex-changer, proper reloading, and thermometer setting at insertion.	Trial run: thermal response test, pipe wall temperature test, exchange well temperature and humidity test, flow test, power test, water pressure test
	Monitoring of fluid in exchange wells	Multipurpose water quality gauge	Liquid in the well		
		Pressure type water gauge	Liquid in the well		
	Real-time monitoring of pipeline structures	Water leak detection service	Liquid pipeline vulnerability point (turning point)		
	Get information on surrounding groundwater	Water leak detection sensor	Groundwater after borehole		

Self-painted by the author.

## Data Availability

The data presented in this study are available in [Automatic Fire Alarm System General Catalog] [Gas detection unit KD-5A/KD-5B Specifications] [Gas detection unit KD-5D/KD-5O Specifications] [Noise and Vibration Level Indicator RTK-27R Specifications] [Catalog: Wireless Mini Logger LR8512, LR8513, LR8514, LR8515, LR8520] [Gas detection and alarm system for shield tunneling works, ditches and tunnels TDL-1/MDL-700 Specifications] [Oxygen and Carbon Monoxide Meter XOC-353II/XOC-353IIBT Specifications] [Digital dust meter (dust meter) LD-5 Specifications] [KYOSAN-Safeguards for Passenger Transfer Area] [NSG-Catalog of Track equipment monitoring-Track material monitoring device/ Orbit displacement monitoring device] [SAKATA DENKI-Catalog of Line equipment monitoring device] [Catalog: Extensometer Z4D-C01] [Catalog: KB-AB/KB-AC (Horizontally tiltometer)] [Catalog: FAC series (Crack gauge)] [Optical Fiber Measurement Solution FBG Sensing] [Kyowa-Measuring Components General Catalog 2021] [Catalog: Crack detection sensor KZCB-A] [Cloud-based IoT Remote Leakage Monitoring System Leaknets Cellular LNL-C Materials].

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
