# Peer review of "Guidelines for Installation of Sensors in Smart Sensing Platforms in Underground Spaces"

_sensors, 2022, doi:10.3390/s22093215_

Round 1
Reviewer 1 Report
In this work, the authors proposed guidelines for sensor installation in underground space platforms. The manuscript is clear and detailed, the authors classified the underground space, analyzed the scene requirements, and sort out the requirements for sensors in the underground space. This article is of interest to better understanding and improving sensor installation in this kind of space platforms. Therefore, it can be considered for publication.
Author Response
Dear Reviewer:
Thank you for raised various problems, we have revised all the issues and responded to the problems. The following is the specific adjustments.
Point 1: The problem of inappropriate research design.
Response 1: On the problem of inappropriate research design, we improved the description of the background, summarized the content of the existing guidelines and summarized the types of sensors used in the current underground space (page 2-3) to support the rationality of our research design.
Point 2: The problem of unclear results presenting.
Response 2: On the problem of unclear results presenting, we added a detailed description of the usage figures (figure.7,9,11,13,15,17) to the original figures to clarify our research results.
Point 3: English language and style are fine/minor spell check required.
Response 2: On the problem of spelling error, we fixed the spelling error in table 4,5,6.
These are the adjustments made for the problems in review. Looking forward to the reply!

Reviewer 2 Report
The paper's objectives are reasonably outlined, centred on analyzing the requirements of different underground space scenes to establish the sensor types and corresponding sensing-scenes parameters. The need for the intervention arises mainly from the authors' perceptions of a lack of design guidelines for setting up intelligent sensing devices in the whole underground space, even if various countries have complete laws and regulations for this kind of infrastructure construction. The foreseen solution is a new set of guidelines of sensor installation for smart scenes, but it is somewhat unclear what is missing in practice, and more details are necessary.
Therefore, the research promises to achieve results that are not very ambitious in terms of innovation overall. It is not entirely clear against what background these features are to be judged as innovative as the description of the current conditions is very generic and fragmented to many issues. As a result, the paper does not convincingly show a novel vision and oscillates between generic statements, despite some innovative methodological elements.
Author Response
Response to Reviewer Comments
Dear Reviewer
Thank you for raising various problems, we have revised all the issues and responded to the problems. The following is the specific adjustments.
Point 1: The need for the intervention arises mainly from the authors' perceptions of a lack of design guidelines for setting up intelligent sensing devices in the whole underground space, even if various countries have complete laws and regulations for this kind of infrastructure construction. The foreseen solution is a new set of guidelines of sensor installation for smart scenes, but it is somewhat unclear what is missing in practice, and more details are necessary.
Response 1: We improved the description of the background, summarized the content of the existing guidelines and summarized the types of sensors used in the current underground space (page 2-3) to support the rationality of our research design.
Point 2: It is not entirely clear against what background these features are to be judged as innovative as the description of the current conditions is very generic and fragmented to many issues. As a result, the paper does not convincingly show a novel vision and oscillates between generic statements, despite some innovative methodological elements.
Response 2: On the problem of not entirely clear against what background these features are to be judged as innovative as the description of the current conditions is very generic and fragmented to many issues, a new table (table 1) have been added to show the limitations of the existing guidelines; a textual summary (page 2) of the current situation is also presented in Introduction to clarify the motivation and necessity of our research; table1 also can be contrasted with table 6 in results to show how is better than existing work.
Point 3: The problem of not providing sufficient research background.
Response 3: On the problem of not providing sufficient research background, in response to research that is not clear what is missing in practice, we improved the description of the background (page 2), we summarized the content of the existing design guidelines in Japan, including the types of equipment mentioned in the guidelines and the details of the equipment requirements mentioned in the guidelines. We found that the existing guidelines lack requirements for specific smart devices as a way to support the necessity of our research.
Point 4: The problem of inappropriate research design.
Response 4: On the problem of inappropriate research design, we summarized the content of the existing guidelines and summarized the types of sensors used in the current underground space (table 1) to support the rationality of our research design.
Point 5: The problem of inadequate description of research methods.
Response 5: On the problem of inadequate description of research methods, we summarized the current equipment used in the underground space (page 2, table 1), and found that the current equipment used in the underground space is also less smart. The collation of the content of the existing guidelines and the collation of the current equipment clarifies the missing content in the current guidelines for underground space design, thus clarifying the viability of our research method.
Point 6: The problem of unclear results presenting.
Response 6: On the problem of unclear results presenting, we added a detailed description of the research design (page 18-22), used to clarify research results;
Point 7: The problem of conclusion not being supported by research results.
Response 7: On the problem of conclusion not being supported by research results, we added a detailed description of the usage figures (figure.7,9,11,13,15,17) to the original Fig. 6-11 to clarify the research results and also support the conclusions. It is also used to demonstrate innovation in usage of the scenarios(interaction between users and scenarios).
These are the adjustments made for the problems. Looking forward to the reply!
Reviewer 3 Report
Review:
In this manuscript, the authors analyzed the requirements of different types of underground space scenes and establish the sensor types and corresponding parameters for smart sensing scenes. According to the sensor parameters, the authors proposed a system design for a smart sensing platform in underground space and form the sensor installation guidelines.
Weakness: The manuscript requires revision based on identified issues present in the proposed work.
- Add related work and compare it. Provide a comparison table with limitations and how to mitigate these limitations with the proposed work.
- Based on the title, motivation is very short in the Introduction, please revise it.
- Improve the quality of Fig. 5 because it is the main part or methodological flow of the proposed research. Redraw and improve.
- According to Figure 6-11, provide proof of how to work these figures. Please explain in detail.
- It is very important that compare to the existing work. And show how is better than existing work.
- The authors should remove the grammatical mistakes and typos in the paper.
- Cross-reference all citations and ensure that they match accordingly. Reference paper format should be uniform. I have identified some of the references with missing details like page numbers, volume numbers, issue numbers, etc. Recent reference must be added, the following is recommended:
- Metal-Organic Frameworks as an Alternative Smart Sensing Platform for Designing Molecularly Imprinted Electrochemical Sensors
- A comprehensive survey on core technologies and services for 5G security: taxonomies, issues, and solutions
- Smart Sensing and End-Users' Behavioral Change in Residential Buildings: An Edge-Based Internet of Energy Perspective
- Analysis of software testing techniques: Theory to practical approach
Author Response
Dear Reviewer,
Thank you for raising various problems, we have revised all the issues and responded to the problems. The following is the specific adjustments.
Point 1: Add related work and compare it. Provide a comparison table with limitations and how to mitigate these limitations with the proposed work.
Response 1: On the problem of add related work and compare it, a new table (table 1) have been added to show the limitations of the existing guidelines to clarify the and necessity of our research; table1 can be contrasted with table 6 in results to show how is better than existing work.
Point 2: Based on the title, motivation is very short in the Introduction, please revise it.
Response 2: We improved the description of the background in the Introduction, a new table (table 1) have been added to show the limitations of the existing guidelines; a textual summary (page 2) of the current situation is also presented in Introduction to clarify the motivation and necessity of our research.
Point 3: Improve the quality of Fig. 5 because it is the main part or methodological flow of the proposed research. Redraw and improve.
Response 3: We have enhanced the image quality of Fig.5 at page 13 and enlarged the font for better readability to clarify our research results.
Point 4: According to Figure 6-11, provide proof of how to work these figures. Please explain in detail.
Response 4: For each image of Fig.6-11, we have added a new image of the smart interaction function to explain how the main sensors mentioned in Fig.6-11. where the changed Fig.6/7 corresponds to the original Fig.6, the changed Fig.8/9 corresponds to the original Fig.7, the changed Fig.10/11 corresponds to the original Fig.8, the changed Fig.12/13 corresponds to the original Fig.9, the changed Fig.14/15 corresponds to the original Fig.10, and the changed Fig.16/17 corresponds to the original Fig.11. It is also used to demonstrate innovation in usage of the scenarios (interaction between users and scenarios).
Point 5: It is very important that compare to the existing work. And show how is better than existing work.
Response 5: A new table (table 1) have been added to show the limitations of the existing guidelines; a textual summary (page 2) of the current situation is also presented in Introduction to clarify the motivation and necessity of our research; table1 also can be contrasted with table 6 in results to show how is better than existing work.
Point 6: The authors should remove the grammatical mistakes and typos in the paper.
Response 6: On the problem of spelling error, we fixed the spelling error in Table 4,5,6.
Point 7: Cross-reference all citations and ensure that they match accordingly. Reference paper format should be uniform. I have identified some of the references with missing details like page numbers, volume numbers, issue numbers, etc. Recent reference must be added, the following is recommended: • Metal-Organic Frameworks as an Alternative Smart Sensing Platform for Designing Molecularly Imprinted Electrochemical Sensorsï¼›• A comprehensive survey on core technologies and services for 5G security: taxonomies, issues, and solutionsï¼›• Smart Sensing and End-Users' Behavioral Change in Residential Buildings: An Edge-Based Internet of Energy Perspectiveï¼›• Analysis of software testing techniques: Theory to practical approach.
Response 7: On the problem of references, the format of the references was clarified and adjusted (references 5,14,24,28,29), inappropriate reference was removed (original references 30), and new references were added to clarify the context and method of the study (references 17,18,31,32).
Point 8: The problem of not providing sufficient research background.
Response 8: On the problem of not providing sufficient research background, in response to research that is not clear what is missing in practice, we improved the description of the background (page 2), we summarized the content of the existing design guidelines in Japan, including the types of equipment mentioned in the guidelines and the details of the equipment requirements mentioned in the guidelines. We found that the existing guidelines lack requirements for specific smart devices as a way to support the necessity of our research.
Point 9: The problem of inappropriate research design.
Response 9: On the problem of inappropriate research design, we summarized the content of the existing guidelines and summarized the types of sensors used in the current underground space (table 1) to support the rationality of our research design.
Point 10: The problem of inadequate description of research methods.
Response 10: On the problem of inadequate description of research methods, we summarized the current equipment used in the underground space (page 2, table 1), and found that the current equipment used in the underground space is also less smart. The collation of the content of the existing guidelines and the collation of the current equipment clarifies the missing content in the current guidelines for underground space design, thus clarifying the viability of our research method.
These are the adjustments made for the problems . Looking forward to the reply!

Round 2
Reviewer 2 Report
The paper could be accepted as it is.